# Towards Australian Regional Turnaround: Insights into Sustainably Accommodating Post-Pandemic Urban Growth in Regional Towns and Cities

**Mirko Guaralda [1], Greg Hearn [1], Marcus Foth [1,\*] , Tan Yigitcanlar [2], Severine Mayere [2] and Lisa Law [3]**

[1] QUT Design Lab, Queensland University of Technology, 2 George Street, Brisbane 4000, Australia; m.guaralda@qut.edu.au (M.G.); g.hearn@qut.edu.au (G.H.)

[2] School of Built Environment, Queensland University of Technology, 2 George Street, Brisbane 4000, Australia; tan.yigitcanlar@qut.edu.au (T.Y.); severine.mayere@qut.edu.au (S.M.)

[3] Division of Tropical Environment and Societies, James Cook University, PO Box 6811, Cairns 4870, Australia; lisa.law@jcu.edu.au

\* Correspondence: m.foth@qut.edu.au; Tel.: +61-7-3138-8772

**Abstract:** The COVID-19 pandemic has made many urban policymakers, planners, and scholars, all around the globe, rethink conventional, neoliberal growth strategies of cities. The trend of rapid urbanization, particularly around capital cities, has been questioned, and alternative growth models and locations have been the subjects of countless discussions. This is particularly the case for the Australian context: The COVID-19 pandemic heightened the debates in urban circles on post-pandemic urban growth strategies and boosting the growth of towns and cities across regional Australia is a popular alternative strategy. While some scholars argue that regional Australia poses an invaluable opportunity for post-pandemic growth by 'taking off the pressure from the capital cities'; others warn us about the risks of growing regional towns and cities without carefully designed national, regional, and local planning, design, and development strategies. Superimposing planning and development policies meant for metropolitan cities could simply result in transferring the ills of capital cities to regions and exacerbate unsustainable development and heightened socioeconomic inequalities. This opinion piece, by keeping both of these perspectives in mind, explores approaches to regional community and economic development of Australia's towns and cities, along with identifying sustainable urban growth locations in the post-pandemic era. It also offers new insights that could help re-shape the policy debate on regional growth and development.

**Keywords:** regional towns; regional cities; regional Australia; regional lifestyle location; regional innovation system; regional turnaround; post-pandemic urban growth; COVID-19 impact; regional planning; sustainable urban development

## 1. Introduction

The ongoing COVID-19 pandemic has affected many countries and millions of people all over the globe. This most severe global health crisis impacted public health, disrupted consolidated neoliberal economies, labor markets, and other facets of social and individual life. Subsequently, the global economy has hit a recession, where it is now gradually turning into a global financial crisis [1]. Today, we are witnessing substantial changes in various social activities and behaviors of individuals, such as reducing economic and social engagements, social distancing, delaying school, staying more at home, working and studying remotely, and moving to safer and less denser locations [2–4].

COVID-19 is a reminder of the vulnerability of major cities as dominant centers of economic and social life; the pandemic has seriously challenged our economic, social, and urban systems and it is still too early to assess its long-term impacts. As Sir Norman Foster has stated [5], our cities will survive, but we need to rethink our model of growth and economic structure. Some cities, so far, have done better than others [6]. However, in Australia, the factors affecting the spread of the virus such as international access points and difficulties of confining spread in geographical clusters, have meant that large cities have become the epicenter [7]. Regional towns and cities may not have had as dramatic health impacts, but certainly their economies have been severely affected in concert with the national economy as a whole. In parallel, the unprecedented move to remote work, nomadic/mobile work, and home-based work has accentuated the use of digital technologies, which obviate the need for geographic co-location to conduct business. These trends have been further accelerated by the pandemic [8–10]. This potentially opens up opportunities for Australia's regional towns and cities, especially for realizing a regional turnaround. Nevertheless, if ever there was a time for innovative approaches to decentralizing Australia's population and economic resources, it is now.

This opinion piece explores and discusses current opportunities for Australian regional towns and cities. The Australian borough of statistics defines towns as settlements with a population between 10,000 and 100,000 [11], but for our purposes we find it helpful to specify that regional towns and cities are those that lie beyond the major capital cities of Sydney, Melbourne, Brisbane, Perth, Adelaide, and Canberra. We follow the Regional Australia Institute's [12] nuanced definitions where regional cities are more than 50,000 people, and towns can include connected lifestyle regions, industry and service hubs, and heartland regions. In other words, these areas are variegated and a one size fits all is not possible. Following this introduction, Section 2 provides background to the regionalism discourses in Australia. Sections 3 and 4 present growth challenges and growth opportunities of Australian regional towns and cities, respectively. Lastly, Section 5 concludes the opinion piece with some remarks on the insights into sustainably accommodating post-pandemic urban growth in regional Australia.

## 2. The Growth of Regional Towns and Cities

In Australia, most of the population is clustered around a few big urban centers (Figure 1). In a country that is almost 90% urban, more than two-thirds of the population are housed by the capital cities and their metropolitan regions. Although relatively few people might live in regional towns, the Regional Australia Institute [12] argues that regional towns and cities collectively house 9.45 million Australians (versus the 15.9 million in the metropolitan capital cities). Moreover, they argue that regional cities and connected lifestyle regions are growing at significant rates (5.4 and 6.8%, respectively) which is comparable to the growth of metropolitan capitals at 8%. In recent decades, like many cities across the world, numerous Australian major cities have been struggling with unprecedented growth, rising costs, and the changing nature of employment [13]. Urban population growth puts pressures on infrastructure, housing, environment, social, and community resilience. The greenfield model of development adopted so far as the prominent strategy to manage urban growth erodes fertile agricultural lands and has a consistent financial impact due to the need of new infrastructures to move people and goods [14].

At the same time, some regional and rural communities fight to keep a sustainable population [15] and experience skills shortages [16], leaving them with the challenge of attracting skilled and motivated workers [17]. Although COVID is projected to slow Australia's population growth [18] and presumably alter these dynamics, the deep contradictions of our cities are the economic and geographic structures that promote 'a winner takes all' urbanism of the talented and advantaged clusters, which succeed in leaving everybody else behind [19]. Population shifts and their urban dynamics already unfold differently in regional towns and cities, where urban landscapes reflect and help reproduce contradictions and inequalities in distinct ways [20]. Nonetheless, even in a century where more than half the world's population is now urban (and rapidly increasing), 'in every place and in every century, there have been alternatives' [21].

A number of planning and development policies are relevant to this discussion. The Regional Australia Institute argues that further agglomeration in capital cities does not improve national economic performance and will impair livability particularly in terms of housing affordability and congestion [22]. They advocate the development of regional settlement strategies, analysis of infrastructure investment options, optimizing land use policies, and better strategies for migration of skilled workers to regional cities. The 'Planning for Australia's Future Population' report also suggests taking the pressure off capital cities, and argues for similar policy interventions, along with an emphasis on well-functioning communities and community services [23]. The Australian Productivity Commission's report on 'Transitioning Australia's Regions' [24] cautions against ad hoc interventions, but also argues for optimizing land use and planning along with rigorous and transparent evaluation of regional development strategies. Their methodology is particularly relevant to our discussion. They developed an index of Regional Adaptive Capacity that emphasizes human, social, natural, financial, and physical capital as well as other economic indicators such as regional economic diversity [24].

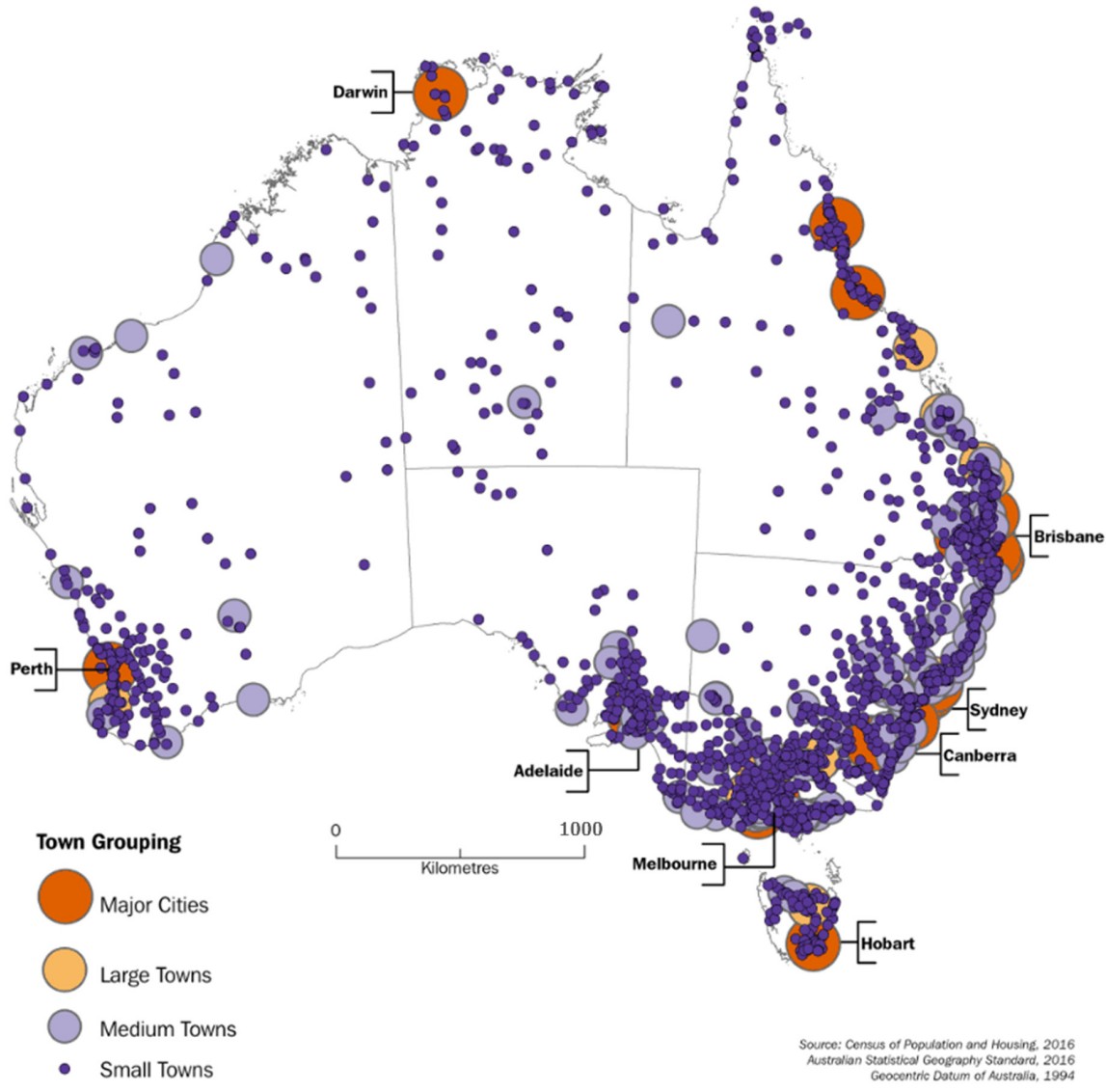

**Figure 1.** Australian settlements by population size groupings [11].

In addition to government-led interventions through policy, we also argue that for these policies to work, regional industries and businesses must have the capacity to innovate and adapt to make use of the opportunities afforded, e.g., better land use and more availability of a high-skilled workforce.

This means continuing transforming industries such as agriculture, manufacturing, and tourism through an emphasis on developing high value-added products and services, as well as accelerating the adoption and tailoring of advanced technology to regional contexts. The emerging technological reality of the modern professional is that an increasingly large number of knowledge workers can choose their desired location and lifestyle while seamlessly collaborating with customers, clients, and stakeholders online [25–27]. These technological forces are doing more than merely shifting economic activities from one area to another. In addition to this geographic movement, these innovations have also been a catalyst for new economic growth and renewal.

Current urban development paradigms replicate at a smaller scale the same pattern used in major cities. The central business district (CBD)-Suburbia model is often implemented with a cookie-cutter approach that does not take into consideration the peculiarities and lifestyle advantages of regional centers, their socioeconomic structure, and the synergies between different types of urban forms [28,29]. Promoting growth in regional towns—a.k.a., medium (1000–50,000 inhabitants) and large (50,000–100,000 inhabitants) towns as per ABS definition [1], as well as regional cities of 50,000 or more—has to be supported by an investigation into suitable urban designs and urban structures to maximize the use of existing infrastructures and capitalize on the local sense of place and community identity [30–33].

Some scholars argue that Australian regional towns and cities pose an invaluable opportunity for post-pandemic growth by 'taking off the pressure from the capital cities' [22,34]. On the other hand, some scholars warn us on the risks of growing regional towns and cities without carefully designed national, regional, and local planning and development strategies, and transferring the ills of capital cities to regions [35,36]—such as unsustainable development and heightened socioeconomic inequalities. As Archer et al. [37] stress, "it is crucial that environmental sustainability, social inclusion, and livability considerations are also included in the deliberations to develop population-based initiatives . . . not only 'How many Australians?' but 'Where will they live?'" (p. 124). Indeed, there is an urgent need for a national discussion about the Australia-wide distribution of urban areas as well as the long-term planning for different settlement patterns [38].

Against this backdrop, the discussion in this opinion piece sets out to explore approaches to regional community and economic development for recovery and growth of Australia's regional towns and cities, along with identifying sustainable growth locations in the post-pandemic era for Australia.

The discussion is guided by three themes:

i.   Attracting population to regional cities;
ii.  Increasing employment in regional cities with a focus on technology and innovation;
iii. Support urban growth in regional cities exploring suitable sustainable urban development models for third-tier centers.

As an opinion piece, we do not aim to provide empirically grounded answers to these themes. Rather, the purpose of this article is to offer a discussion informed by a critical review of the relevant literature with a view to stimulate additional voices to chime in and further the discourse in the field.

## 3. The Challenges

While, arguably, regional Australia faces many challenges, within the scope of this article we have selected three interrelated issues to focus on. They are: (Section 3.1) The population counter-trend of an undercurrent of people leaving metropolitan cities and migrating and settling in regional areas—a.k.a. counter-urbanization—and the challenge of retaining them and providing career pathways for them; (Section 3.2) the role and impact of digital technology for a sustainable approach to population shifts, settlement strategies, and regionalization; and (Section 3.3) the challenge of reconciling population growth aspirations with protecting lifestyle benefits and a sustainable natural environment.

## 3.1. Offering Sea-Change, e-Change or Flee-Change Locations

The phenomenon of urbanism increasingly impacts on Australian society through population growth in urban areas or decline in rural and regional areas [39]. Population growth puts pressures on infrastructure, housing [40], environment, social networks, and communities. Like cities all over the world, Australian cities struggle with unprecedented growth, rising costs, the changing nature of work—particularly Sydney, Melbourne, and Brisbane [13], and uncertainty in a complex political arena [19]. Economic geographies and associated policies tend to focus on urban concentrations. Yet, there is a lack of attention on realizing significant opportunities for the development of robust and sustainable knowledge and innovation economies in regional areas [41,42]. We note that not all regional cities have the same characteristics; for instance, Gold Coast and Sunshine Coast have been experiencing rapid growth. Yet, the growth faced by metropolitan areas, the shrinking population of the regional centers, which often try to compete in an increasingly globalized economy, and the impact of lifestyle shifters can disrupt the delicate balance of sensitive coastlines and other key environmental features [43]. The impacts are global game changes that threaten Australia's productivity, social cohesion, and livability as we know it. Nonetheless, urbanization has never been either inevitable or without countervailing tendencies—in other words, path dependency matters. However, in every place and in every century, there have been alternatives [21].

Traditionally, coastal cities of Australia have been the locations for 'sea-change' [44]. However, Gurran and Blakely [45] argue coastal migration is incidental to urban dominance, and unlikely to be a factor in the prosperity of Australia's regional cities located outside metropolitan areas. In recent years, as part of a counter-urbanization trend, 'e-change' is underway in Australia—migrating from the capital cities to nearby lifestyle towns [46] and working remotely [4,47]. The latest trend is 'flee-change'. People particularly from VIC and NSW have started to move to coastal cities/towns to escape COVID-19 risks and strict pandemic restrictions [48]. Nevertheless, regional cities outside metropolitan areas, away from the coast and metropolitan CBDs, have been less likely to attract large migrant populations. It is unclear if COVID-19 and the wider embracement of new technology will alter this.

## 3.2. Offering Broadband Access, Innovation, and the Digital Economy Conditions

There is an acknowledged overlap between creative industries, other knowledge work, and remote work, and an established trend for them to seek out lifestyle opportunities [4,49–51]. The connectivity promised by Australia's National Broadband Network (NBN) has also opened up possibilities for regional centers to market themselves as lifestyle towns for telework—also linked with the above-discussed e-change [46,52,53]. Yet, there are benefits and challenges to location choices [54]. There are pressures on some of these lifestyle choices such as the rapid population increases—e.g., Gold Coast, QLD [21]. Nevertheless, the contrasting and highly criticized dystopian effect of high population growth on the Gold Coast raises issues of urban and transport planning and design policy questions for the promotion and growth of sensitive and sustainable regional centers [21,55].

Within the digital economy, knowledge and creative occupations—and digital nomadism—are important [56] and have been growing with each census since 2001 [57,58]. Lobo et al. [59] argue that higher densities of educated creative and knowledge workers spread innovations throughout the economy and impact metropolitan productivity broadly. A number of studies have documented the dynamics of knowledge and creative work outside CBD areas where they are strongest [60–63]. In addition to the knowledge and creative industries, there may be a general 'employment dividend' by embedding knowledge and creative workers [64] in other larger sectors such as manufacturing [57,58,65,66].

Regional employment opportunities through innovation are critical. Some of the success stories include the Ideas Lab in Cairns (formerly the Cairns Innovation Centre)—a self-sustaining start-up and innovation ecosystem in Far North Queensland [67]—and Australian Tropical Sciences and Innovation Precinct—a world class tropical research hub located in Townsville [68]. Nonetheless, both projects are

public/academic sector investments, and there is only a negligible level of private sector driven large investments in regional Australia—excluding mining and agricultural operations. The lack of private sector led innovation activities in regional cities and the lack of effective partnerships pose a risk for the adoption of knowledge and innovation economy [69,70].

### 3.3. Offering Regional Liveability and Sustainability

The concepts of livability and sustainability are increasingly used to define our cities [71]. Pressures on metropolitan urban areas are creating vast disparities between inner suburbanites and the outer regions of a city. Consequently, while this crisis is often reported as a statistical phenomenon, it is an urban reality for millions of individuals, their lives, their work, and the livability of their neighborhood. As housing prices soar in metropolitan cities of Australia, there is a generation of youth questioning if they will ever be able to enter the housing market and an older generation trying to maintain home ownership under increasing economic pressure and uncertainty [72]. Besides, commuting robs families and communities of time and social input, undermining the social structures of our society—especially in the case of Sydney and Melbourne. At the same time, many regional and rural communities fight to keep a sustainable population [15] and experience skills shortages [16], leaving them with the challenge of attracting skilled and motivated workers.

This issue brings the importance of developing and delivering new urban planning, place-based design, and placemaking strategies tailored to regional Australia [73–76] that avoid regional cities and towns from being propelled to join their larger counterparts in the new urban crisis. This also assists in addressing one of the practical research challenges identified by the Commonwealth government of developing more resilient urban, rural, and regional infrastructure. Without a desired level of livability and sustainability, regional urban offerings are unlikely to become a growth alternative to the primate—i.e., Sydney—and second-tier cities of Australia—other state capitals and some cities within their metropolitan area [77].

## 4. The Opportunities

In response to the triad of challenges we identified and presented above, we now turn our attention to a similarly interrelated set of three areas of opportunities for regional Australia. They are: (Section 4.1) The policy response; (Section 4.2) the planning and design response, and (Section 4.3) the governance response.

### 4.1. The Need for a Distinctive Policy for Regional Knowledge and Innovation Industry Dynamics

In line with the international focus on the knowledge and innovation industries as a source of potential cultural and economic innovation and growth, numerous regional areas have identified the stimulation of knowledge and innovation industries as a key element for sustainable growth [78]. Further, knowledge and innovation industries and knowledge workers are not restricted to urban areas: These industries exist as a site of strong potential innovation and development beyond major cities [79]. In fact, many knowledge workers opt to move themselves and their work away from urban centers. Evidence signals that many knowledge workers are attracted to specific regional areas as part of a wider 'e-change' ex-urban migration trend [54]. However, there is a gap in research on knowledge/creative industries in regional locations [80].

This issue brings the following questions to mind: (a) How do knowledge and innovation industries work in regional locations? (b) What can knowledge and innovation industries and workers be attracted to regional locations? (c) In what types of places do regional knowledge and innovation industries thrive best? (d) How is technology—specifically the shift to telecommuting—playing a role in shaping regional innovation cultures and practices? There is limited research conducted on these questions, which signals that knowledge and innovation industries in regional locations do not operate in the same ways as they do in the CBDs of first- and second-tier cities [81,82].

Understanding how Australian regional knowledge industries operate on the ground could provide a foundation for regional-place specific innovation policies and initiatives in Australia. This could also inform the development of policies and programs that avoid replicating metropolitan urban strategies, which may well miss the mark. Hence, uncovering the economic and cultural geographies of knowledge industries innovation in regional Australia is critical to provide a strong basis for evidence-based policymaking.

*4.2. The Policy, Planning, and Design Response*

The literature highlights the impact of placemaking and the built environment on the livability of regional centers and their ability to become globally competitive as places for people to live and work [83]. It is, hence, imperative to explore the factors that make the biggest and most pertinent impacts, supporting knowledge industries and workers and their spatial needs, and develop urban planning and design policy recommendations tailored to a digital economy in regional Australia. Urban planning and design strategies adopted regionally are often simply pale imitations of metropolitan approaches that do not resonate with local communities [84]. Given e-changers and flee-changers are moving to regional centers (mostly coastal) to avoid overly crowded and congested urban milieus—along with others with different relocation reasons—it is important to work closely with local communities to deliver novel urban planning and design strategies that are engineered to maximize and protect the local identity and capitalize on local opportunities. In that perspective, it is useful to adopt innovative participatory action research and participatory design methodologies to address this component [85,86].

*4.3. The Governance Response*

Enabling local governments in regional Australia to navigate the challenges outlined here will require a balance of impact, lifestyle, and shifting economies. Achieving this balance will in turn require a concerted and collective effort to foster innovation, strong policy direction, and an understanding of the key drivers and influences. Issues of pollution, inequalities, rising housing costs and affordability, cost of living, transport networks and capacities, community and social connection place pressures on lifestyle and health and, in doing so, compromise the livability for the masses [87–89]. Florida [19] suggests the 'new urban crisis' is not just a crisis of our cities but of our age, of a highly urbanized knowledge-based capitalism. Cities have increasingly become a patchwork of concentrated advantage and much larger swathes of disadvantage. Yet, this patchwork model is also a faithful reflection of the methodological approaches that have emerged in the literature for addressing this suite of challenges. These topics have been separately examined through the lenses of pollution [90], economics [91,92], and social science [93], but prior investigations in this area have so far failed to provide any kind of integrative analysis that draws together these disparate bits into a more cohesive, unified perspective that can guide local governance. In sum, strategic governance highly matters for further development of Australian regional cities [94], and effective urban planning, design, policy, and monitoring—along with the planner as "an orchestrator and enabler of planning regional futures" [95]—are the key vehicles to achieve the desired outcomes and regional futures.

## 5. Concluding Remarks

The COVID-19 pandemic has forced us to rethink our cities from the prism of health, livability, and sustainability [96,97]. Referring to climate change and COVID-19, Bauman [98] argues that, "the existential emergencies we face require a wholesale reimagining of how we live, work and play in urban spaces". Specifically in the context of the post-pandemic recovery driven by a renewed attention to regional Australia, we argue that this will require a continuing focus on food and agriculture [99,100], as well as radically reframing our unhealthy relationship with nature and the planet towards a post-anthropocentric design agenda [72,101] that prioritizes revegetation, rewilding, and more-than-human perspectives [102–104].

The pandemic has also created an opportunity to revisit our options on human settlement, urban habitat, and where and how to locate the future population (and also economic) growth in Australia [105]. We have a chance to rethink the structure of our urban systems and pursue alternative models to the consolidated paradigms of Euclidean planning. This opinion piece underlined the following regional challenges to attract and retain migrants to regional cities (particularly coastal ones): (a) Offering sea-change, e-change, and flee-change locations; (b) offering broadband access, innovation, and the digital economy conditions; and (c) offering regional livability and sustainability. It also highlighted the crucial importance of the following mechanisms for sustainably accommodating the post-pandemic urban growth in regional Australia: (a) A distinctive policy for regional knowledge and innovation industry dynamics; (b) the planning and design response; and (c) the governance response.

Our enquiry has revealed that the three original themes used to structure our discussion can be articulated in a series of diverse and detailed questions, to further the understanding of opportunities and dynamics in regional towns:

- Attracting population to regional cities: (a) What factors affect the choice of location in moving to regional Australia? (b) What placemaking strategies and urban design features are necessary and desired to enable knowledge workers to work effectively in Australian regional lifestyle locations? (c) How can the unique local qualities and the 'sense of place' of regional Australia be maintained, while catering to the demands and needs of new workers' families? (d) How can we attract skilled migrants to regional Australia through cultural interventions? (e) What are the new models of affordable housing for regional Australia? (f) How can the pandemic migrants be retained in regional Australia post pandemic?
- Increasing employment in regional cities with a focus on technology and innovation: (a) How can existing businesses grow their revenues and jobs? (b) What specific skills are needed in regional Australia? (c) How can creative and knowledge services grow revenues of existing businesses in regional Australia? (d) What models of start-up and innovation hubs are best suited to regional Australia? (e) How can the community, industry, and local government sectors come together to spur innovation in regional Australia? (f) What are the profiles of jobs, professions, and education that are most likely to choose and succeed in relocating to regional Australia? (g) How can the economic development strategies (including land use policies) in regional Australia best respond to these insights? (h) What is the prospect of new service providers and business models emerging to support innovation across regional Australia? (i) How can we create new and sustainable industries in regional Australia?
- Support urban growth in regional cities exploring suitable sustainable urban development models for third-tier centers: (a) What models of cities can optimize the sustainable growth in regional Australia? (b) What is the role of land use policy and developmental regulations in accelerating the creation of innovation precincts/districts in regional Australia? (c) How can innovation/coworking hubs be designed to enable affordable workspace and access to equipment, and knowledge networks in regional Australia? (d) How can more inclusive and diverse innovation ecosystems be developed in regional Australia? (e) How can urban redevelopment be rethought to better suit the context and opportunities of regional Australia? (f) How can these changes impact the economic prospects and prosperity for regional Australia?

We conclude this opinion piece—that explores approaches to regional community and economic development of Australia's regional towns and cities (Australian regional turnaround), along with providing sustainable urban growth locations in the post-pandemic era—by restating the perspective by Archer et al. [5]: We need "through better evidence, to ensure that distributing growth to regional areas achieves its proper place in the policy debate and that government agencies invest the time and effort required to make good decisions on these opportunities" (p. 35). The themes highlighted in this piece can contribute to re-shaping the policy debate on regional growth and development and identify new avenues for research and practice.

**Author Contributions:** Conceptualization, formal analysis, and original draft preparation; G.H. and M.F.; formal analysis, and writing—original draft preparation, T.Y.; formal analysis, and writing—review and editing, M.G., S.M., and L.L. All authors have read and agreed to the published version of the manuscript.

**Funding:** This research received no external funding.

**Acknowledgments:** This research did not receive any specific grant from funding agencies in the public, commercial or not-for-profit sectors. The authors thank Nick Osbaldiston of James Cook University, and three anonymous referees for their invaluable comments on an earlier version of the manuscript.

**Conflicts of Interest:** The authors declare no conflict of interest.

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
