# Peer review of "Towards Australian Regional Turnaround: Insights into Sustainably Accommodating Post-Pandemic Urban Growth in Regional Towns and Cities"

_sustainability, doi:10.3390/su122410492_

Round 1
Reviewer 1 Report
The paper is a clear and concise overview integrating many aspects of regional city growth. My only concern is that there is some over-reach in that the aim in the introduction that the paper explores 'transformative approaches' doesn't align as well with the findings and conclusion as it could do. Some tempering of the initial aims would be helpful as the conclusions drawn are quite suitable given the informing research material referenced. So toning down the aims to reflect the conclusions would I think strengthen the paper. The conclusions drawn, for example, in section 4 that more consultative approaches need to be used and that strategic governance is important are quite sensible, but the case is not made for them to be transformative - ie how would these changed practices transform the trajectories for regional cities that have been set out in sections 3 and 4?
In particular there is a long list of questions set out between lines 141 and 169, and most of these are not addressed in the discussion presented under the 3 main themes. In my view the 3 main themes already form a useful framework, and including a long list of un-addressed questions feels like it weakens the otherwise clear narrative of the paper. It would need to be a much longer paper to address all the questions raised, and I don't think this is the point of the paper. So I would suggest some rewriting of this to either distil the questions down to those that are addressed in the paper, or to indicate that there are indeed a great many questions worthy of (future) consideration (by the research community) which are beyond the scope of this paper.
Reviewer 2 Report
It is an opinion article. I found it interesting, but some aspects are not clearly addressed:
- it is not clear in what sense is the proposed approach "transformative"? It should be more exactly explained.
- on page 3, r. 106 it should be specified which policies are meant?
- on page 4, r. 140 it is not clear according to which criteria have the authors defined third-tier cities?
- Figure 1 should indicate distinctive third-tier cities.
- it is not clear how the three themes indicated on page 4 as a collection of questions relate to the rest of the paper?
Reviewer 3 Report
The topic of this paper is consistent with the theme of this journal. The researchers explore sustainable urban growth in Australia. Please find enclosed my remarks and suggestions for your guidance.
Overall, I think that this is a good paper. The abstract could include another sentence to describe the implications of this research.
The authors could have clarified their research question of their contribution. They can also explain how their paper builds on the extant literature and/or addresses a knowledge gap in academia.
The conclusions could clearly specify the implications of this contribution.
I am recommending minor revisions to improve the quality of this paper. Best wishes.
I suggest that you submit your contribution(s) to this special issue: https://www.mdpi.com/journal/sustainability/special_issues/Smart_Cities_Digital_Innovation
